# Association of SNP rs5069 in *APOA1* with Benign Breast Diseases in a Mexican Population

**DOI:** 10.3390/genes13050738

**Published:** 2022-04-22

**Authors:** Carolina Domínguez-Díaz, María Cristina Morán-Moguel, Rosa Elena Navarro-Hernandez, Rebeca Romo-Vázquez, Adriana Patricia Mendizabal-Ruiz

**Affiliations:** 1Doctoral Program in Biomedical Sciences, Physiology Department, CUCS, Universidad de Guadalajara, Jalisco 44340, Mexico; carolina.dominguezd@alumnos.udg.mx; 2Department of Molecular Biology and Genomics, CUCS, Universidad de Guadalajara, Jalisco 44340, Mexico; cmmoguel@yahoo.com.mx; 3Immunometabolism in Complex Diseases and Aging, Department of Molecular Biology and Genomics, CUCS, Universidad de Guadalajara, Jalisco 44340, Mexico; rosaelena.navarro@cucs.udg.mx; 4Traslational Bioengineering Department, CUCEI, Universidad de Guadalajara, Jalisco 44430, Mexico; rebeca.romo@academicos.udg.mx; 5Pharmacobiology Department, Universidad de Guadalajara, CUCEI, Jalisco 44430, Mexico

**Keywords:** apolipoproteins, breast cancer, benign breast disease (BBD), Mexico, single nucleotide polymorphisms, rs5069

## Abstract

Breast cancer (BCa) is the most common type of cancer affecting women worldwide. Some histological subtypes of benign breast disease (BBD) are considered risk factors for developing BCa. Single nucleotide polymorphisms (SNPs) in the genes encoding apolipoproteins A-I (*APOA1*) and B (*APOB*) have been associated with BCa in Tunisian, Chinese, and Taiwanese populations. The objective of this pilot study is to evaluate the possible contribution of *APOA1* and *APOB* polymorphisms to BCa and BBD in the Mexican population. We analyzed the association of 4 SNPs in genes encoding apolipoproteins: rs670 and rs5069 in the *APOA1* gene, and rs693 and rs1042031 in the *APOB* gene, by performing PCR-RFLP with DNA extracted from the biopsy tissue of Mexican women with BCa or BBD and whole blood samples obtained from the general population (GP). Our results showed an association between the CT + TT genotypes of the SNP rs5069 and BBD (*p* = 0.03201). In the A-T haplotype, the frequency of the SNPs rs670 and rs5069 differed significantly between the BBD group and the GP and BCa groups (*p* = 0.004111; *p* = 0.01303). In conclusion, the SNP rs5069 is associated with BBD but not with BCa in the Mexican population.

## 1. Introduction

Breast cancer (BCa) is the most common female cancer worldwide. In Latin America, it is the leading cause of female cancer-related mortality [1]; 37% of women in Mexico have BCa [2].

A relationship between obesity and cancer has been reported, and abnormal lipid metabolism is implicated in carcinogenesis mechanisms [3,4]. Lipids are small hydrophobic or amphipathic molecules that serve as energy sources and play a key role as signaling molecules and regulators of various cellular processes, including proliferation, differentiation, signaling, and metabolism [5,6]. The deregulation of these processes in cancer cells shows that lipids are essential in promoting cancer cell growth and survival. The connection between altered lipid levels and the cancer process suggests that lipid profiles can be used to diagnose and assess disease progression [6]. Lipid biomarkers are expressed differently between healthy subjects and patients with breast tumors and can help distinguish between benign and malignant lesions [7,8].

Apolipoproteins play a crucial role in lipid metabolism and cancer development, and the polymorphisms in their genes influence lipid abnormalities related to cancer [9]. High-Density Lipoproteins (HDL), Low-Density Lipoproteins (LDL), Apolipoprotein A-I (ApoA-I), and Apolipoprotein B-100 (ApoB-100) levels are associated with BCa development and severity [10,11,12,13,14,15]. Also of interest is their role in benign breast diseases (BBD), a heterogeneous group of non-malignant lesions [16] associated with lipid control variation, with BBD patients having higher HDL and lower LDL levels than healthy controls and BCa patients, respectively [17,18]. Both BCa and BBD patients have high levels of serum ApoA-I [19]. In addition, proliferative BBD increases the risk of developing in situ BCa up to 5-fold, depending on proliferation and atypia [16,20,21,22,23], making BBD an important risk factor for BCa development.

ApoA-I is the major protein constituent of HDL and is involved in cholesterol transport from peripheral tissues to the liver [24,25]. The single nucleotide polymorphism (SNP) rs670 involves an A->G substitution at 75 bp upstream of the *APOA1* transcription initiation site [26]. The A allele is associated with BCa risk in Tunisian and Taiwanese populations [24,27]. SNP rs5069 in *APOA1* replaces a C>T in intron 1 [26], and its CT genotype is associated with BCa lymph node spread in the Tunisian population [24]. Both rs670 and rs5069 affect the expression of ApoA-I. The A allele of rs670 and the CT genotype of rs5069 are associated with higher levels of this apolipoprotein [24,28,29].

ApoB-100 is a part of VLDL (Very Low-Density Lipoproteins), IDL (Intermediate Density Lipoproteins), LDL, and chylomicrons [30]. SNP rs693 is a silent C->T substitution in *APOB* exon 26, while SNP rs1042031 changes a G->A in exon 29, resulting in glutamate instead of lysine. These SNPs have an opposite role in ApoB-100 expression and cholesterol metabolism. The T allele of rs693 increases the expression of this apolipoprotein, while the change of amino acid provoked by rs1042031 disrupts its binding to the LDL receptor. These variants were associated with BCa risk in a Chinese population [31]. Although SNPs in *APOA1* and *APOB* are associated with BCa, none of the studies have assessed both genes in BCa or BBD. In this pilot study, we investigated the association between *APOA1* and *APOB* SNPs and breast neoplasms in a Mexican population. Our results showed a positive association between *APOA1* SNPs and BBD risk, rather than with BCa.

## 2. Materials and Methods

Samples of unrelated individuals without a family history of BCa and without prior treatment were selected from a collection of 109 genomic DNA samples extracted from paraffin-embedded neoplastic tissues studies previously [32,33]. Samples genotyped for all four SNPs from two genes (*APOA1*-rs670, *APOA1*-rs5069, *APOB*-rs693, *APOB*-rs1042031) were sorted according to the diagnosis, with 39 in the BCa group, and 19 in the BBD group (Appendix A), and were compared to 150 peripheral blood genomic DNA samples from nonrelated and healthy women of ages 18 to 72 years from a general population (GP), analyzed previously [34]. Informed consent was obtained from the subjects. The Institutional Ethics Committee of CIBO-IMSS approved this study.

SNP assays were performed using PCR-RFLP. Appendix A lists the reagents, experimental conditions, and the length of amplified fragments. We designed all primers except for MSP1 [35]. The rs670 and rs5069 regions were amplified using MSP1 and MSP2 primers. Each *APOA1* site was also amplified with pairs MSP1-MSP3 and MSP2-MSP4 for rs670 and rs5069, respectively. SNPs in *APOB* were amplified with the primers listed in Appendix A. 

Statistical analyses were performed with PLINK v1.07 [36]. The genotypic association was tested with χ^2^ test. Fisher’s exact test was used for the Hardy–Weinberg equilibrium (HWE) calculation in each group, and the association model tests, and *p*-values were adjusted using false discovery rate (FDR). Linkage disequilibrium (LD) was calculated using the squared correlation coefficient based on genotypic allele counts. Haplotype-specific tests were performed for all SNPs. Results with *p* ≤ 0.05 were considered significant.

## 3. Results and Discussion

To test the association of *APOA1* and *APOB* SNPs with BCa and BBD, samples from breast tumor biopsies were genotyped for rs670, rs5069, rs693, and rs1042031. From the 109 DNA samples initially considered in the archive collection, only 58 samples could be genotyped for all four polymorphisms. Table 1 lists these SNPs’ genotype and allele distributions in our sample groups. 

The *APOA1* SNPs rs670 and rs5069 were in HWE in all our groups, and only their genotype distributions differed significantly between the BBD and GP groups (*p* = 0.0197 and *p* = 0.0454, respectively, Table 1). The rs670 A allele was associated with BCa [24,27] and renal cancer [37] in previous studies. In our study, the rs670 variant did not show any association with the BCa group; however, the frequencies of this allele and its AA genotype were significantly different between the BBD and GP groups (*p* = 0.0335 and *p* = 0.0268, respectively, Table 2). After the FDR correction, no differences were noted. Further studies are needed to characterize *APOA1* variants in non-malignant breast pathologies.

Our analysis showed a significant FDR-corrected association between the TT + CT genotypes and BBD (*p* = 0.032, Table 2). After independently testing the TT and CT genotypes with the CC genotype as a reference, only the CT genotype presented an association with BBD (*p* = 0.0116, Appendix A). Interestingly, in our study, the CT genotype frequencies observed in BCa were lower than in BBD. These frequency distributions could be a reflection of the increased ApoA-I levels associated with this genotype. The evidence for apolipoproteins and HDL roles in cancer is mounting, yet some results are conflicting, and many questions remain. In terms of ApoA-I levels, some studies of ApoA-I expression in BCa development report a direct association, while for others, it is inverse [9]. An assay in neoplastic mice reported that an elevated ApoA-I/HDL ratio protects against tumor development; moreover, after tumor establishment, some of these mice were infused with Human ApoA-I, leading to tumor shrinkage [14]. Another venue of studies has tried to relate ApoA-I genotypes with HDL cholesterol and serum ApoA-I levels with mixed results. Most studies have not shown any significant genotype [29,38,39,40,41,42]. Those with a significant result associate the CT genotype with higher levels of both [28,43,44,45]. The only exception was a Spanish population that related the T allele with lower HDL-cholesterol levels [46]. Since cancer progression is a non-linear multistage event where mutations are not always additive toward malignancy and heterogeneous genotypes present an advantage against several diseases, we hypothesize that the CT genotype increases BBD risk in Mexican populations while at the same time lowering BCa risk. Elucidating the role of *APOA1* in BCa progression warrants further exploration

Liu et al. [31] analyzed a BCa population by BMI and found an overrepresentation of rs693 CT + TT genotypes and rs1042031 GA + AA genotypes in overweight and obese individuals, respectively, despite statistical insignificance. In our results, rs1042031 deviated from HWE in the GP group and rs693 in the BCa group (*p* = 0.011 and *p* = 0.0181, respectively). None of these SNPs showed an association. The equilibrium skewness of rs693 in the BCa group suggested either a cancer-based genetic drift or LD with another cancer-promoting locus; however, the small sample size in our study hindered further analysis. As previously reported [31], we also found that the rs1042031 GA + AA genotypes were overrepresented in overweight individuals in the GP group. The results persisted after excluding this subgroup from the analysis. Our case groups could not be tested for any BMI-based genotype trend because such information was unavailable. Cancer studies should evaluate BMI since *APOB* SNPs probably have shifting relevance in cancer, depending on an individual’s BMI.

In LD analysis, we observed low values for r^2^, indicating weak SNP pairwise cor-relations (Table 3). Haplotype frequency distributions (Table 4) show that the H1 haplotype (A-T) was significantly different between the BBD and the GP (*p* = 0.0041) and between the BCa and the BBD groups (*p* = 0.01303). These results are consistent with the genotype analysis results of rs670 and rs5069, in that the rs670 A and rs5069 T alleles are associated with BBD in the Mexican population. Because we didn’t find an association with BCa, we surmise that these alleles could promote benign proliferative changes in breast tissue and have a potential role as tissue biomarkers to identify benign breast diseases. However, as the small sample size limits our ability to interpret the results, future studies are needed to determine its prognostic value further. The H4 haplotype frequency (G-C) differs significantly between the BBD and GP groups (*p* = 0.0111). Given its low frequency in BBD, it could affect cancer initiation and progression, warranting further research. 

Notably, in our study, GP genomic DNA was obtained from peripheral blood, while in BCa and BBD patients, DNA was obtained from breast biopsies, which may present with loss of heterozygosity, potentially influencing local lipid-immune metabolism pathways that induce carcinogenicity. This could be tested by analyzing germline genotypes derived from non-tumoral tissue or blood cells, but this data was unavailable for our groups. Despite such limitations, this is the first pilot study to examine the interactions of rs670, rs5069, rs693, and rs1042031 and their association with BCa and BBD.

## 4. Conclusions

Our results, while limited due to the sample size, show that the CT + TT genotypes of rs5069 and the H1 haplotype due to the rs670 A allele and rs5069 T allele are associated with BBD. While further studies are necessary, our pilot study suggests that rs670 and rs5069 could increase benign tumor risk in Mexican women.

## Figures and Tables

**Table 1 genes-13-00738-t001:** Genotype and allelic distribution in the studied groups.

SNP	Genotypes/Alleles	BCa *n* = 39 (%)	BBD *n* = 19 (%)	GP *n* = 150 (%)	Genotypic Association Test *^a^*
*APOA1* rs670	GG	12 (30.8)	5 (26.3)	63 (42.0)	*p* = 0.2274 BCa vs. BBD
	GA	22 (56.4)	8 (42.1)	70 (46.7)	*p* = 0.439 BCa vs. GP
	AA	5 (12.8)	6 (31.6)	17 (11.3)	*p* = 0.0454 * BBD vs. GP
	**G**	46 (59)	18 (47.4)	196 (65.3)	
	A	32 (41)	20 (52.6)	104 (34.7)	
*APOA1* rs5069	CC	27 (69.2)	8 (42.1)	110 (73.3)	*p* = 0.119 BCa vs. BBD
	CT	10 (25.6)	10 (52.6)	37 (24.7)	*p* = 0.541 BCa vs. GP
	TT	2 (5.1)	1 (5.3)	3 (2.0)	*p* = 0.0197 * BBD vs. GP
	**C**	64 (82.1)	26 (68.4)	257 (85.7)	
	T	14 (17.9)	12 (31.6)	43 (14.3)	
*APOB* rs693	CC	11 (28.2)	5 (26.3)	52 (34.7)	*p* = 0.3955 BCa vs. BBD
	CT	26 (66.7)	11 (57.9)	78 (52.0)	*p* = 0.1838 BCa vs. GP
	TT	2 (5.1)	3 (15.8)	20 (13.3)	*p* = 0.7654 BBD vs. GP
	**C**	48 (61.5)	21 (55.3)	182 (60.7)	
	T	30 (38.5)	17 (44.7)	118 (39.3)	
*APOB* rs1042031	GG	26 (66.7)	17 (89.5)	115 (76.7)	*p* = 0.1693 BCa vs. BBD
	GA	12 (30.8)	2 (10.5)	28 (18.7)	*p* = 0.2369 BCa vs. GP
	AA	1 (2.6)	0 (0)	7 (4.7)	*p* = 0.3925 BCa vs. GP
	**G**	64 (82.1)	36 (94.7)	258 (86.0)	
	A	14 (17.9)	2 (5.3)	42 (14.0)	

*^a^*: *p*-value calculated using χ^2^ test of association for all the three genotypes. The wild-type allele of each SNP is shown in bold-faced letters. BCa: Breast cancer group; BBD: Benign Breast disease group; GP: General population group. * *p* < 0.05.

**Table 2 genes-13-00738-t002:** rs670, rs5069, rs693 and rs1042031 association analysis under different genetic models for all groups.

SNP (Gene)	Genetic Model	BCa vs. GP			BBD vs. GP			BCa vs. BBD		
		OR	*p*	*p*-Corr	OR	*p*	*p*-Corr	OR	*p*	*p*-Corr
rs670 (***APOA1***)	AA + GA vs. GG	1.63	0.2703	0.5406	2.03	0.2222	0.3372	0.80	1	1
	AA vs. GG + GA	1.15	0.7823	0.7823	3.61	0.0268 *	0.0803	0.32	0.1506	0.3011
	A vs. G	1.31	0.354	0.6363	2.09	0.0335 *	0.0670	0.63	0.32	0.4267
rs5069 (***APOA1***)	TT + CT vs. CC	1.22	0.6878	0.6878	3.78	0.0080 **	0.0320 *	0.32	0.0847	0.2168
	TT vs. CC + CT	2.65	0.2748	0.4122	2.72	0.3822	0.5733	0.97	1	1
	T vs. C	1.31	0.4772	0.6363	2.76	0.0170 *	0.0670	0.47	0.1534	0.3067
rs693 (***APOB***)	TT + CT vs. CC	1.35	0.5678	0.6878	1.48	0.6092	0.6092	0.91	1	1
	TT vs. CC + CT	0.35	0.2596	0.4122	1.22	0.7265	0.7265	0.29	0.3179	0.3179
	T vs. C	0.96	1	1	1.25	0.5986	0.5986	0.77	0.5502	0.5502
rs1042031 (***APOB***)	AA + GA vs. GG	1.64	0.2186	0.5406	0.39	0.2529	0.3372	4.25	0.1084	0.2168
	AA vs. GG + GA	0.54	1	1	0	1	1	-	1	1
	A vs. G	1.34	0.3752	0.6363	0.34	0.1980	0.2640	3.94	0.0854	0.3067

OR = Odds ratio. Fisher’s exact test was used to determine significant differences between groups. *p*-Corr is the adjustment of *p*-values after FDR correction. BCa: Breast cancer group; BBD: Benign Breast disease group; GP: General population group. * *p* < 0.05; ** *p* < 0.01.

**Table 3 genes-13-00738-t003:** Linkage disequilibrium analysis for general population, breast cancer, and benign breast disease groups.

SNP A	SNP B	r^2^		
		GP	BCa	BBD
rs670	rs5069	0.002	0.0986	0.0266
rs693	rs1042031	0.1841	0.0049	0.0032

SNP A refers to the SNP in the first locus, and SNP B refers to the SNP on the second locus. A moderate LD can be considered when r^2^ > 0.2. BCa: Breast cancer group; BBD: Benign Breast disease group; GP: General population group.

**Table 4 genes-13-00738-t004:** Haplotype association analysis in breast cancer, benign breast disease, and general population groups.

Haplotype		BCa Frequency	GP Frequency	χ^2^	*p*
rs670|rs5069					
**H1**	AT	0.0404	0.0478	0.076	0.7828
**H2**	GT	0.1391	0.0955	1.252	0.2631
**H3**	AC	0.3698	0.2989	1.448	0.2289
**H4**	GC	0.4507	0.5578	2.855	0.09109
rs693|rs1042031					
**H6**	CA	0.1535	0.1381	0.1179	0.7314
**H7**	TG	0.3651	0.392	0.1839	0.6681
**H8**	CG	0.4814	0.4699	0.0320	0.8581
**Haplotype**		**BBD Frequency**	**GP Frequency**	**χ^2^**	** *p* **
rs670|rs5069					
**H1**	AT	0.2007	0.0659	8.234	0.0041 **
**H2**	GT	0.1151	0.0774	0.6363	0.4251
**H3**	AC	0.3256	0.2808	0.3318	0.5646
**H4**	GC	0.3586	0.5759	6.437	0.0112 *
rs693|rs1042031					
**H6**	CA	0.0526	0.14	2.274	0.1316
**H7**	TG	0.4474	0.3933	0.4105	0.5217
**H8**	CG	0.5	0.4667	0.1504	0.6981
**Haplotype**		**BCa Frequency**	**BBD Frequency**	**χ^2^**	** *p* **
rs670|rs5069					
**H1**	AT	0.0381	0.173	6.164	0.0130 *
**H2**	GT	0.1414	0.1428	0.0004	0.9846
**H3**	AC	0.3722	0.3533	0.0394	0.8427
**H4**	GC	0.4483	0.3309	1.456	0.2276
rs693|rs1042031					
**H5**	TA	0.0567	0.0212	0.7458	0.3878
**H6**	CA	0.1228	0.0314	2.535	0.1114
**H7**	TG	0.3279	0.4261	1.069	0.3011
**H8**	CG	0.4926	0.5212	0.0839	0.772

BCa: Breast cancer group; BBD: Benign Breast disease group; GP: General population group. * *p* < 0.05; ** *p* < 0.01.

## Data Availability

Not applicable.

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
