# Peer review of "Association of SNP rs5069 in APOA1 with Benign Breast Diseases in a Mexican Population"

_genes, 2022, doi:10.3390/genes13050738_

Round 1

Reviewer 1 Report

The manuscript is well written and easy to follow. It reports interesting findings on the association of common genetic variants at the APOAI and APOB gene locus with breast cancer risk. Such studies are limited and therefore this brief report would add to the limited knowledge on the role of genetics variants at gene loci involved in lipid transport with increased/decreased risk for breast cancer.

  1. In the introduction, define VLDL when first mentioned.
  2. The authors should elaborate on the lipid metabolism between BC and BBD and if implicated to other forms of cancer.
  3. Indicate the “rs” number for the APOAI polymorphisms I table S2.
  4. The HWE for the studied variants has to be clearly stated. The authors mentioned that APOB SNPs deviated from HWE. How about the SNPs for APOAI, were they in HWE?
  5. Elaborate on how “the CT genotype increases BBD risk while against cancerous cell lines.”
  6. The authors should consider reporting LD before the haplotype analysis.
  7. In table 3, check the third row as I think the comparison was for BDD vs GP. The authors should check this and correct accordingly.
  8. The authors should take into account and consider the limitations of the small sample size and consider their study as a pilot study.

Author Response

Comment 1. In the Introduction, define VLDL when first mentioned.

Response: We agree with this and have added the meaning of VLDL and IDL to the manuscript on page 2, lines 66 – 67.

Comment 2. The authors should elaborate on the lipid metabolism between BC and BBD and if implicated to other forms of cancer.

Response: Thank you for pointing this out. We agree with this; therefore, we have added the following paragraph to the Introduction to explain the relationship between lipid metabolism and breast tumors (page 1, lines 38 – 46).

"Lipids are small hydrophobic or amphipathic molecules that serve as energy sources and play a key role as signaling molecules and regulators of various cellular processes, including proliferation, differentiation, signaling, and metabolism [5,6]. The deregulation of these processes in cancer cells shows that lipids are essential in promoting cancer cell growth and survival. The connection between altered lipid levels and the cancer process suggests that lipid profiles can be used to diagnose and assess the disease progression [6]. Lipid biomarkers are expressed differently between healthy subjects and patients with breast tumors and can help distinguish between benign and malignant lesions [7,8]".

Comment 3: Indicate the "rs" number for the APOAI polymorphisms I table S2.

Response: Agree. We have modified Table S2 to add the rs number of the APOAI polymorphisms with their respective primers.

Comment 4:The HWE for the studied variants has to be clearly stated. The authors mentioned that APOB SNPs deviated from HWE. How about the SNPs for APOAI, were they in HWE?

Response: Indeed, the APOA1 SNPS were in HWE. Here we list the p value obtained for rs670 and rs5069 in our groups:

rs670:

GP p= 0.8569

BCa p= 0.5062

BBD p= 0.645

rs5069:

GP p= 1

BCa p=0.5775

BBD p= 0.6069

To clarify this, we have added to the manuscript a line (line 109) stating this: "The APOA1 SNPs rs670 and rs5069 were in HWE in all our groups."

We have also specified which APOB SNPs deviated in HWE for our GP and BCa groups: "In our results, rs1042031 deviated from HWE in the GP group and rs693 in the BCa group (p=0.011 and p=0.0181, respectively)" (lines 145 – 147.)

 Comment 5: Elaborate on how "the CT genotype increases BBD risk while against cancerous cell lines."

Response: Thank you for this suggestion. Accordingly, we have rewritten the paragraph to clarify the idea while stating it as a hypothesis derived from our results.

 Comment 6: The authors should consider reporting LD before the haplotype analysis.

Response: Agree. We have rewritten the text to report LD analysis first, instead of the haplotype analysis, on page 5, lines 156 – 157. We have also changed the order of Tables 3 and 4 to reflect this change in the text.

 Comment 7: In table 3, check the third row as I think the comparison was for BDD vs GP. The authors should check this and correct accordingly.

Response: Thank you for pointing this out. Indeed the comparison is for BBD and GP. We have corrected the name of the column of Table 3 on page 5.

Comment 8: The authors should take into account and consider the limitations of the small sample size and consider their study as a pilot study.

Response: Thanks for the observation and suggestion. We have rewritten the manuscript to reflect this change and mention that this is a pilot study. The changes can be found on the abstract, page 1, line 21; Introduction, page 2, line 74; discussion, page 6, line 184; and conclusions, page 6, line 189.

Reviewer 2 Report

Domínguez-Díaz and collogues in “ ASSOCIATION OF SNP rs5069 IN APOA1 WITH BENIGN  BREAST DISEASES IN A MEXICAN POPULATION” evaluated 4 SNPs that associated with breast cancer and breast begin breast disease.

I have some concerns below:

  • As the study’s aim was to use the specific SNP loci as the clinical biomarkers for following treatment, then it demands large datasets to support their research and analysis results. Current study design using 39 BCa + 19 BBD for the comparison obviously not enough for the analysis, and also from the results, we could see after the FDR correction, the adjusted p value of main important findings getting no significant.
  • The authors should include much more datasets for the validation with their findings.
  • The authors cannot get such kind of conclusion as this study just have 19 patients from BBD and most of statistics have not reached significant level: “This confirms the association of rs670 and rs5069 with BBD in the Mexican population, suggesting that both rs670 A and rs5069 T alleles influence breast cell proliferation and, since it is not associated with BCa, these alleles have a potential role in becoming tissue biomarkers to identify benign breast diseases.”
  • The last section of Introduction should not include the conclusions in lines 56-58.
  • How many targeted loci for the SNP assays when introducing SNP genotypes in the first and second paragraphs in “Materials and Methods” section?
  • Utilized softwares should be noted with the version in the main text.
  • The results and discussion section mixed with introduction in the first paragraph in lines 80-84.
  • In addition, the results section was largely mixed with discussion and make it hard to follow which results are from this study that distinguish from previous research.
  • The sentence was quite confusing “The rs670 A allele, associated with BCa [15,18] and renal cancer [31], did not show an association with BCa”
  • The result did not show in any table “After independently testing the TT and CT genotypes with the CC genotype as reference, only the CT genotype presented an association with BBD (p=0.0116)”.

Author Response

Comment 1: As the study's aim was to use the specific SNP loci as the clinical biomarkers for following treatment, then it demands large datasets to support their research and analysis results. Current study design using 39 BCa + 19 BBD for the comparison obviously not enough for the analysis, and also from the results, we could see after the FDR correction, the adjusted p value of main important findings getting no significant.The authors should include much more datasets for the validation with their findings.

Response: Thank you for this suggestion. The main problem with our sampling was using a pathology embedded tissue bank, where many samples from our original cohort had to be excluded for several reasons. However, we cannot extend the sample size because we did not receive approval to test new patients. Although this study is limited by its small sample size, we are advancing to publish what we believe are interesting findings. While the results from this study should be considered preliminary, they have the potential to contribute to future analyses. Considering the other reviewer's suggestion, we report this research as a pilot study, and we have rewritten our findings to reflect our limitations.

 Comment 2: The authors cannot get such kind of conclusion as this study just have 19 patients from BBD and most of statistics have not reached significant level: "This confirms the association of rs670 and rs5069 with BBD in the Mexican population, suggesting that both rs670 A and rs5069 T alleles influence breast cell proliferation and, since it is not associated with BCa, these alleles have a potential role in becoming tissue biomarkers to identify benign breast diseases."

Response: Thank you for pointing this out. In consideration of our limitations due to the small sample size, we have rewritten this statement as follows: "These results are consistent with the genotype analysis results of rs670 and rs5069, that the rs670 A and rs5069 T allele are associated with BBD in the Mexican population. Because we did not find an association with BCa we surmise that these alleles could promote benign proliferative changes in breast tissue and have a potential role as tissue biomarkers to identify benign breast diseases. However, as the small sample size limits our ability to interpret the results, future studies will be needed to determine its prognostic value further".

This change can be found on page 5, lines 159 – 165.

Comment 3: The last section of Introduction should not include the conclusions in lines 56-58.

Response: Agree. We have, accordingly, removed these lines from the Introduction.

Comment 4: How many targeted loci for the SNP assays when introducing SNP genotypes in the first and second paragraphs in "Materials and Methods" section?

Response: Thank you for pointing this out. We have incorporated this information into the first paragraph in the Material and Methods section, on page 2, lines 81-82.

 Comment 5: Utilized softwares should be noted with the version in the main text.

Response: Agree. we added the software version used on page 2, line 93.

 Comment 6: The results and discussion section mixed with Introduction in the first paragraph in lines 80-84.

Response: Agree. we have removed this paragraph from the results and discussion section and added it to the Introduction on page 2, lines 47 – 48, and lines 55 – 57.

Comment 7: In addition, the results section was largely mixed with discussion and make it hard to follow which results are from this study that distinguish from previous research.

Response: We consider that the best way to present our results is as a brief report. For this reason, we have combined the results and discussion sections. We have rewritten and modified our manuscript to differentiate our results from previous studies.

Comment 8: The sentence was quite confusing "The rs670 A allele, associated with BCa [15,18] and renal cancer [31], did not show an association with BCa"

Response: We agree with this and have rewritten the sentence to clarify that although some studies report an association of this allele with breast and renal cancer, our results did not show any association with BCa group (lines 110 – 111).

Comment 9: The result did not show in any table "After independently testing the TT and CT genotypes with the CC genotype as reference, only the CT genotype presented an association with BBD (p=0.0116)".

Response: Thank you for pointing this out. We have added a new table on page 3 of the Supplementary Information of this manuscript to incorporate the results obtained from all the independent tests of each genotype with their corresponding reference genotype for all SNPs. Also, we added this reference to the manuscript on page 3, line 119

Reviewer 3 Report

Carolina Dominguez-Diaz and group have studies correlation between APOA1 and APOB apolipoproteins with breast cancer and benign breast disease. Their study shows that CT+TT genotypes of rs5069 and the H1 haplotype due to rs670 158 A allele and rs5069 T allele are associated with BBD. Their study is interesting however it is just correlative.  Please provide answers for following queries and discuss them in manuscript appropriately.

  1. Table 1 – Why general population has SNPs present for APOA1 and APOB?
  2. Table 1 – Please mention wildtype and SNP nucleotide residues separately in table itself.
  3. If SNPs are present in general population, then are they at risk of developing breast cancer or BBD?
  4. There are multiple SNPs present in APOA and APOB genes, then why only rs670, rs5069, rs693, and rs1042031 were chosen for study?
  5. How does these rs670, rs5069, rs693, and rs1042031 SNPs affect function of respective apolipoproteins? Please discuss.

Author Response

Comment 1: Table 1 – Why general population has SNPs present for APOA1 and APOB?

Response: As we used a general population as our control for this study, and not just healthy subjects, we can find the distribution of all possible genotypes, regardless of the effect of a particular allele.

 Comment 2: Table 1 – Please mention wildtype and SNP nucleotide residues separately in table itself.

Response: Thank you for your suggestion. We have used bold letters to distinguish the wildtype nucleotide of each SNP and indicated each wildtype genotype in Table 1.

 Comment 3:  If SNPs are present in general population, then are they at risk of developing breast cancer or BBD?

Response: While our study is limited due to the small sample size, the results suggest that the SNPs could increase BBD risk. However, it is important to note that other factors could influence if a person develops BBD.

 Comment 4: There are multiple SNPs present in APOA and APOB genes, then why only rs670, rs5069, rs693, and rs1042031 were chosen for study?

Response: The SNPs rs670, rs5069, rs693, and rs1042031 are among the most commonly studied variants in APOA and APOB. These SNPs have been studied due to their association with altered lipid levels in several lipid-related diseases. The SNPs rs670 and rs5069 of APOA1 are associated with higher circulating Apo-AI protein levels [10.1016/j.ijcard.2004.10.017; 10.1186/s13000-015-0328-7]. SNPs rs693 and rs1042031 are associated with higher levels of LDL-C and disruption of the binding between LDL-C and its receptor [10.1093/jjco/hyt018\rhyt018]. Since altered lipid profiles have been associated with cancer, we chose to analyze these polymorphisms that appear to affect the levels of this apolipoprotein. Furthermore, some genotypes of these polymorphisms were associated with cancer in Chinese, Tunisian, and Taiwanese populations. None of them have ever been evaluated on the Mexican population. Likewise, they have not been evaluated in benign breast lesions, another group of diseases also related to alterations in lipid levels. For this reason, we used DNA obtained both from patients with BCa and BBD to analyze these 4 SNPs and their association with these pathologies.

 Comment 5: How does these rs670, rs5069, rs693, and rs1042031 SNPs affect function of respective apolipoproteins? Please discuss.

Response: Thank you for this suggestion. We have added this information to the Introduction, on page 2, lines 61 – 72. 

Round 2

Reviewer 2 Report

In addtion to the sample size limit, the authors solved my other concerns. 

Reviewer 3 Report

authors have satisfactorily answered my queries. I still feel novelty factor of manuscript is low. However findings from this manuscript can be beneficial to scientific community, so I am recommending this article for publication.